# Alpha-Synuclein Preformed Fibrils Induce Cellular Senescence in Parkinson’s Disease Models

**DOI:** 10.3390/cells10071694

**Published:** 2021-07-05

**Authors:** Dinesh Kumar Verma, Bo Am Seo, Anurupa Ghosh, Shi-Xun Ma, Karina Hernandez-Quijada, Julie K. Andersen, Han Seok Ko, Yong-Hwan Kim

**Affiliations:** 1Neuroscience Program, Department of Biological Sciences, Delaware State University, Dover, DE 19901, USA; dverma@desu.edu (D.K.V.); aghosh@desu.edu (A.G.); karina.h521@gmail.com (K.H.-Q.); 2Department of Neurology, School of Medicine, The Johns Hopkins University, Baltimore, MD 21205, USA; bseo5@jhmi.edu (B.A.S.); shixun625@jhmi.edu (S.-X.M.); 3Neuroregeneration & Stem Cell Program, Institute for Cell Engineering, School of Medicine, The Johns Hopkins University, Baltimore, MD 21205, USA; 4Buck Institute for Research on Aging, Novato, CA 94945, USA; jandersen@buckinstitute.org

**Keywords:** cellular senescence, alpha-synuclein preformed fibrils, SATB1, Lamin B1, HMGB1, p21, reactive astrocytes, microglia activation

## Abstract

Emerging evidence indicates that cellular senescence could be a critical inducing factor for aging-associated neurodegenerative disorders. However, the involvement of cellular senescence remains unclear in Parkinson’s disease (PD). To determine this, we assessed the effects of α-synuclein preformed fibrils (α-syn PFF) or 1-methyl-4-phenylpyridinium (MPP^+^) on changes in cellular senescence markers, employing α-syn PFF treated-dopaminergic N27 cells, primary cortical neurons, astrocytes and microglia and α-syn PFF-injected mouse brain tissues, as well as human PD patient brains. Our results demonstrate that α-syn PFF-induced toxicity reduces the levels of Lamin B1 and HMGB1, both established markers of cellular senescence, in correlation with an increase in the levels of p21, a cell cycle-arrester and senescence marker, in both reactive astrocytes and microglia in mouse brains. Using Western blot and immunohistochemistry, we found these cellular senescence markers in reactive astrocytes as indicated by enlarged cell bodies within GFAP-positive cells and Iba1-positive activated microglia in α-syn PFF injected mouse brains. These results indicate that PFF-induced pathology could lead to astrocyte and/or microglia senescence in PD brains, which may contribute to neuropathology in this model. Targeting senescent cells using senolytics could therefore constitute a viable therapeutic option for the treatment of PD.

## 1. Introduction

Neurodegenerative disorders including Alzheimer’s disease (AD) and Parkinson’s disease (PD) are caused by aging-related neuronal loss [1,2]. PD is characterized by selective dopaminergic neuronal cell death within the substantia nigra pars compacta (SNpc), which is derived from an alpha-synuclein-mediated intraneuronal protein aggregation known as Lewy body formation [3]. Although PD is a well-defined neurodegenerative disease, the contribution of cellular senescence to dopaminergic neuronal loss is not yet well studied. Cellular senescence is considered a potent tumor suppressive mechanism resulting in cell cycle arrest which triggers a pro-inflammatory response known as the senescence-associated secretory phenotype or SASP [4,5]. Since neurons are non-proliferative, they have traditionally not been considered to be capable of undergoing cellular senescence, although excessive extra- or intra-cellular stress can lead to neuronal death as well as cellular senescence [5,6]. Recent publications have suggested that senescence and the SASP may be strong contributing factors to PD pathology [4,6,7,8]. However, it is still not well understood what cell-types undergo stress-mediated senescence in the central nervous system (CNS).

Conventionally, protein misfolding-mediated protein aggregation is considered a critical target for halting neuronal cell death associated with PD [1,2]. However, despite numerous attempts, approaches that interrupt the formation of protein aggregation have yet been proven to be successful in preventing neurodegeneration [9,10]. There is mounting evidence, however, that the presence of senescent cells in the CNS contributes to the neuropathology associated with neurodegenerative diseases, possibly even after protein aggregates are removed [9,10,11,12]. Therefore, it is plausible that neurodegeneration is, at least in part, mediated by additional mechanisms including the induction of cellular senescence [5,9,12]. To assess the potential for proteotoxic stress to act as an inducer of cellular senescence in the context of PD, we interrogated the effects of α-synuclein pre-formed fibrils (α-syn PFF)-mediated toxicity in mouse brain and ex vivo and in vitro models of the disorder [13,14]. We further examined whether the markers of cellular senescence were associated with human PD pathology in SNpc. We hypothesized that α-syn aggregates trigger stress-induced premature senescence (SIPS) and/or activate astrocytes and microglia [5,6,10,15]. To address this, the presence of cellular senescence was determined using well-known senescence markers, including Special AT-rich sequence binding protein 1 (SATB1), Lamin B1 and High Mobility Group Box 1 (HMGB1), tumor suppressors p21 and p16, and the endo/lysosomal marker, LAMP1 [16,17,18,19,20], as well as abnormal astrocyte morphology in α-syn PFF-induced PD mouse models [8,21,22]. Our data show that α-syn PFF treatment reduced the levels of SATB1, LaminB1 and HMGB1 and led to an increase in the levels of p21 [7,20]. Since SATB1, HMGB1 and Lamin B1 are chromatin organizer/transcription factor, chromatin protein or nuclear membrane proteins, respectively, the regulation of those proteins would be involved in gene expression, particularly related to cellular senescence [5,6]. Our results strongly suggest that the cellular senescence process plays a role in PD pathology through senescence induction in astrocytes and possibly microglia in brain regions affected in PD [8,21,22]. Based on these findings, ridding the brain of senescent astrocytes or “astrosenescence” via senolytic interventions could constitute a more feasible therapeutic option than conventional protein aggregation-based prevention [5,9,15,16,22,23,24].

## 2. Material and Methods

### 2.1. Animals

All animal protocols were conducted in accordance with the United States Public Health Service Guide for the Care and Use of Laboratory Animals; all procedures were approved by the Institutional Animal Care and Use Committee (IACUC) at Delaware State University or Johns Hopkins, School of Medicine. Four or five animals per polyacrylic cage were housed with access to food and water ad libitum and were maintained in standard housing conditions, i.e., room temperature 22 ± 1 °C and humidity 60–65% with 12:12 light:dark cycle. Until mice were anesthetized for the experiments, the number of animals and the level of discomfort were minimized for the experiments [21,25].

### 2.2. Cell Culture and Cellular Fractionation with α-Syn PFF Treatment

The N27 parental cell line was obtained from EMD Millipore (SCC048, Burlington, MA, USA), used only under 20 passage number and maintained in RPMI 1640 supplemented with 10% FBS and 1% Penicillin-Streptomycin at 37 °C and 5% CO_2_ using standard cell culture methods [25,26]. MPP^+^ (1-methyl-4-phenylpyridinium) was given at the concentration of 640 μM for 24 h. Cells were also treated with 1 µg/mL of endotoxin-free mouse α-syn PFF (SPR-324, StressMarq, Victoria, BC, Canada) or PBS for 6, 12, 24 and 48 h, and then lysed in RIPA buffer (50 mM Tris, 100 nM NaCl, 1% NP-40, 1 mM NaF, 2 mM Na_3_VO_4_, 1 mM PMSF and 1% protease phosphatase inhibitors), and sonicated briefly on ice for lysis (3 s on and 10 s off for 5 cycles) for Western blots. Cell lysates were centrifuged at 15,000 *g* for 10 min, and collected supernatants were transferred to fresh vials and stored at −20 °C or used immediately, as previously reported [25].

For cellular fractionation, 3 × 10^5^ N27 cells/well were seeded into 6-well plates. On the following day, cells were treated with 5 μg/mL of preincubated Wheat Germ Agglutinin (WGA, Sigma, St. Louis, MO, USA), followed by incubation with media containing α-syn PFF (1 µg/mL, StressMarq) and GlcNAC (0.1 M, Sigma, St. Louis, MO, USA) for 48 h. Cytosolic and nuclear fractions were isolated by using NE-PER kit (Cat. 78833; Thermo, Waltham, MA, USA), per the manufacturer’s instructions. Briefly, cells were harvested with 0.1% trypsin-EDTA and then centrifuged at 500 *g* for 5 min. Cells were washed by resuspending cell pellets in PBS. Ice-cold CER I buffer (100 µL) was added to the cell pellet and then fractionation was performed according to the manufacturer’s instructions.

### 2.3. α-Synuclein Purification and α-Syn PFF Preparation

Recombinant mouse α-synuclein proteins were purified using an IPTG independent inducible pRK172 vector system as previously described [21]. Endotoxin was depleted with an endotoxin removal kit (ToxinEraser, Genscript, Piscataway, NJ, USA). For primary cell culture (Figures 4–6) and tissue extraction (Figure 8), α-syn PFF (5 mg/mL) was prepared in PBS and then stirred with a magnetic bar (1000 rpm) at 37 °C. After a week of incubation of α-synuclein proteins, aggregates were diluted into 0.1 mg/mL in PBS and then sonicated for 30 s (0.5 s pulse on/off) at 10% amplitude (Branson Digital sonifier, Danbury, CT, USA) [27]. α-Syn PFF was stored at −80 °C until use.

### 2.4. Primary Cortical Neuron Culture and α-Syn PFF Treatment

For primary cell culture, timed pregnant female CD1 mice were purchased from Charles River Laboratories and C57BL/6J male/female mice from Jackson Laboratory. Primary cortical neurons were prepared as previously described [27,28]. Briefly, a single-cell suspension was obtained from embryonic day 16 (E16) pups using timed pregnant female CD1 mice. Mixed cells containing both neurons and glia were then grown on 6-well plates coated with poly-L-lysine, and neurobasal media were supplemented with B27. The cultures were treated with 5 µM cytosine β-d-arabinofuranoside (AraC, Sigma, #C6645, St. Louis, MO, USA) at day of in vitro (DIV) 3 to remove the glial cells [28]. Primary cortical neurons (7 DIV) were treated with 5 µg/mL of endotoxin-free PFFs for 7 days. At 12 h, 24 h, 48 h, 72 h or 7 days after PFF treatment, cortical neurons were washed with ACSF 3 times and extracted in the lysis buffer containing 0.2% SDS, 1% Triton X-100 and 0.5 mM EGTA/EDTA with proteinase/phosphatase inhibitor cocktail for Western blot analysis.

### 2.5. Primary Astrocytic Culture

Astrocytes were cultured in serum-free astrocyte growth media to avoid activation by serum. The prepared single-cell suspension as mentioned above was applied for EasySep Mouse CD11b positive selection to remove microglia before seeding. Astrocyte-rich fraction was seeded in 6-well plates, coated with poly-L-lysine previously and cultured in serum-free base medium containing 50% DMEM, 50% neurobasal, 1 × SATO (recommended by the Cold spring harbor protocol, PMID:18471889 [29]), 100 μg/mL streptomycin, 100 U/mL penicillin, 292 μg/mL l-glutamine, 1 mM sodium pyruvate and 5 μg/mL of N-acetyl cysteine. This medium was supplemented with the astrocyte-required survival factor HBEGF (Peprotech, 100–47, Cranbury, NJ, USA) at 5 ng/mL with a complete medium change on day 2, as previously described [30].

### 2.6. Primary Microglia Culture

Microglia were purified by immunopanning from postnatal day 1 (P1) forebrain and cultured as previously described [21]. Briefly, after removing the meninges, the brains were washed with PBS and then transferred to 0.25% trypsin-EDTA, followed by gentle agitation for 30 min. The complete media, DMEM/F12 (Gibco, 11320033, Waltham, MA, USA) supplemented with 10% heat-inactivated FBS, 50 U/mL penicillin, 50 μg/mL streptomycin, 2 mM l-glutamine, 100 μM non-essential amino acids and 2 mM sodium pyruvate, were used to stop the trypsinization reaction. A single-cell suspension was obtained by trituration through pipetting. Cell debris and aggregates were removed by passing the single-cell suspension using a 100 μm nylon mesh. The prepared single-cell suspension was cultured in T175 flask for 13 days with complete media change on day 6. Microglia were separated using the EasySep Mouse CD11b Positive Selection kit (StemCell, #18970, Cambridge, MA, USA). The magnetically separated microglia-rich fraction was seeded on 6-well plates with complete media.

### 2.7. Human SNpc Tissue Extraction

Human brain samples were obtained via Dr. Yoon-Seung Kim at the Rutgers University, Medical School, NJ from NIH NeuroBioBank (request #197). Eight human PD SNpc post-mortem (n = 8) and age- and gender-matched control tissues (n = 8) were homogenized using glass tissue homogenizer (Cole Palmer, Vernon Hills, IL, USA) in RIPA lysis buffer (50 mM Tris-HCl (pH 7.4), 100 mM NaCl, 1 mM EDTA, 1% NP-40, 1 mM NaF, 2 mM Na_3_VO_4_, 10 mM NEM, 1 mM PMSF, 1% protease phosphatase inhibitor cocktail (Sigma-Aldrich, St. Louis, MO, USA). After homogenization, samples were incubated on ice for 30 min, vortexed at full speed for 15–20 s after every 10 min for complete lysis and then centrifuged at 14,000 *g* for 15 min. The supernatants were collected for further analysis, and the protein levels were quantified using Pierce Rapid Gold BCA Protein Assay kit (Thermo, Waltham, MA, USA).

### 2.8. Mouse Brain Tissue Lysate Preparation

For stereotaxic injection of α-syn PFF (Figure 8), two month-old male and female mice were anesthetized with xylazene and ketamine. An injection cannula (26.5 gauge) was stereotaxically inserted into the striatum (mediolateral, 2.0 mm from bregma; anteroposterior, 0.2 mm; dorsoventral, 2.6 mm) unilaterally (inserted into the right hemisphere). Five to six months after the injection, almost 8-month-old α-syn PFF treated mice were sacrificed for Western blots. Brain tissues were region-specifically isolated and homogenized in lysis buffer with a complete protease inhibitor mixture (Roche, Indianapolis, IN, USA) using a Diax 900 homogenizer (Sigma-Aldrich, St. Louis, MO, USA). After homogenization, samples were rotated at 4 °C for 30 min for complete lysis and then the homogenates were centrifuged at 15,000 *g* for 20 min. The supernatants were collected for further analysis. Protein levels were quantified using the BCA assay (Thermo, Waltham, MA, USA) and 10–20 μg each was separated on SDS-PAGE gels as described [27].

### 2.9. Stereotaxic α-Syn PFF Injection and Brain Isolation

All C57Bl/6 mice were 8+ months old when we received them from Charles River Lab (Wilmington, MA, USA). In the vehicle group, the same volume of 0.9% saline was injected (n = 10 for each). For stereotaxic injection of mouse α-syn PFF (SPR-324, StressMarq) for immunohistochemistry (Figures 9–12), 12-month-old male and female mice were anesthetized with isoflurane [21,25]. An injection cannula (26.5 gauge) was stereotaxically inserted into the dorsal striatum bilaterally [13,14]. The infusion was performed at a rate of 0.2 μL/min, and 2 μL of α-syn PFF (2.5 μg/μL in PBS) or the same volume of PBS. Five to six months following α-syn PFF or PBS injection, 17–18 month-old α-syn PFF-injected mice were deeply anesthetized via isoflurane followed by intra-cardiac perfusion using ice cold 0.9% saline, followed by 150 mM NaCl/70% ethanol. After decapitation, whole brain was isolated. Half of the brain was used for isolation of the striatum and the midbrain and stored at −80 °C for molecular/biochemical analyses, and the other half brain was post-fixed in 4% paraformaldehyde for immunohistochemical analyses [25].

### 2.10. Western Blot Analyses

After protein concentration was determined, protein samples were loaded on precast polyacrylamide gel (SurePAGE Bis-Tris, 4–20%, 12 wells; GenScript, Piscataway, NJ, USA) and transferred to PVDF membrane (Immobilon-P, EMD-Millipore, Burlington, MA, USA) using a Bio-Rad transfer apparatus. Membranes were incubated with specific primary antibodies, such as anti-Lamin B1 (1:1000; MA1-06103, Invitrogen, Waltham, MA, USA), anti-HMGB1 (1:1000; MA5-17278, Invitrogen), anti-SATB1 (1:1000; sc-376096, Santa Cruz, Dallas, TX, USA), anti-p16 (1:1000; sc-1661, Santa Cruz), anti-p21 (1:1000; sc-6246, Santa Cruz), anti-beta III-tubulin (1:500; PA5-25655, Invitrogen), anti-GFAP (1:1000; PA1-10019, Invitrogen), anti-Iba-1 (1:1000; PA5-27436, Invitrogen), anti-Histone3 antibody (1:1000, 50-173-1414, Proteintech, Rosemont, IL, USA) or anti-GAPDH antibody (1:5000; AM4300, Thermo, Waltham, MA, USA) at 4 °C overnight. On the following day, after washing membranes three times in TBS-T, membranes were incubated with secondary anti-mouse (31430, Invitrogen) or anti-rabbit (31460, Invitrogen) IgG antibodies conjugated with HRP (1:10,000) at room temperature for 2 h. The PageRuler pre-stained protein ladder (Thermo 26616) was used to estimate protein molecular weights. Signals were developed in Immobilon Forte Western HRP substrate (EMD-Millipore) and detected under ChemiDoc iBright CL1000 (Invitrogen). The mean intensity (intensity/area) of bands was determined and normalized by GAPDH using ImageJ software (NIH, Bethesda, MD, USA). Immunoblot images were converted into 8-bit gray-scale images and avoided over-saturation. Equal areas corresponding to selected lanes were analyzed on each blot image [25]. Splicing was implemented only for clarity purposes and the adjustment was performed using the original larger image.

### 2.11. Immunohistochemistry

The OCT mounted mouse brains were sectioned using HM525 cryostat (Thermo Scientific, Waltham, MA, USA) throughout the striatum and substantia nigra and mounted onto positively charged slides (Midwest Sci, Valley Park, MO, USA). Sectioned slides were thawed at room temperature for 30 min before processing. Slides were fixed with cold fixative 100% ice-cold acetone for 10 min and then dried at room temperature for 5 min, followed by washing in 0.1 M PB for 10 min, twice. Permeabilization was carried out using permeabilization buffer containing 1% goat serum and 0.4% Triton X-100 in PB (PB-T) twice for 10 min each. After blocking with 5% BSA in PB-T at RT for 30 min, slides were incubated with anti-Lamin B1 (1:500; MA1-06103, Invitrogen), anti-HMGB1 (1:500; MA5-17278, Invitrogen), anti-p21 (1:500; sc-6246, Santa Cruz), anti-GFAP (1:1,000; PA1-10019, Invitrogen), anti-beta III-tubulin (1:500; PA5-25655, Invitrogen) or anti-Iba-1 antibody (1:500; PA5-27436, Invitrogen) at 4 °C overnight. Slides were washed in PB-T three times for 10 min each, followed by incubating with Alexa-488 or -647 conjugated secondary antibodies (1:1000; A11008 or A21235, Molecular Probes, Thermo) at RT for 1 h and washed in PB-T 3 times for 10 min each. As a negative control, the primary antibody step was omitted. Images were first captured under different fluorescent filters, corelating to the secondary antibody used for a particular protein. Coverslips were mounted on slides using DAPI-ProLong Diamond antifade mounting medium (30 µL/slide, P36966, Thermo). Images were acquired using the EVOS microscope (Invitrogen) and processed for maximum intensity projections using Zeiss ZEN software, followed by intensity analysis using ImageJ. For individual protein analysis, each image was divided into six equal square sections, and the fluorescence intensity of each senescence marker co-labeled with a cell-type marker (GFAP, Iba-1 or β-III-tubulin) was measured in a blinded manner. Background was subtracted from each intensity measurement, and total intensity was averaged for the image [31]. For the astrocyte volume measurement, randomly selected GFAP positive cells were outlined using the free-hand drawing tool from ImageJ [21]. Same images were also used to count all the GFAP positive cells to calculate the number of astrocytes in the striatum (STR), Substantia Nigra compacta (SNc) and cortex (CTX) regions. The average was then calculated for individual sections belonging to PBS or PFF treated group. Each group included 5–6 animals (n = 5 for PBS and n = 6 for α-syn PFF). At least 6–8 sections from the region of interest were used for the IHC analyses. Protein fluorescence intensities, astrocyte volume and activated astrocyte numbers were measured by raters who are unaware of the treatment group [25,32].

### 2.12. Statistical Analyses

In most statistical analyses, unpaired Student’s *t*-test was applied to assess PFF or 1-methyl-4-phenylpyridinium (MPP^+^) effects on the expression of cellular senescence markers, compared to vehicle treatment (Figures 1, 3, 7–12 and Appendix A). One-way ANOVA, Tukey’s post-hoc test was applied to Figures 2 and 4–6. The relative intensities of PFF-treated brains were compared with vehicle-treated mice (control, 100%) in Figures 9–12. For all studies, *p* < 0.05 was considered statistically significant (*); GraphPad Prism 8.03 software was used for all data analyses and display. Values are presented as mean ± standard error of the mean (SEM).

## 3. Results

### 3.1. Cellular Senescence Markers Are Induced by Either MPP^+^ or α-Syn PFF Treatment in Dopaminergic N27 Cells

Initially, we assessed the effects of MPP^+^ exposure on cellular senescence markers in rat dopaminergic N27 cells. Cellular senescence markers including SATB1, Lamin B1, high mobility group box 1 (HMGB1) and p16 and p21 were quantified in Western blots (Figure 1A). MPP^+^ treatment was found to reduce the levels of Lamin B1 (Figure 1A,B) and HMGB1 (Figure 1A,C), whereas both p16 (Figure 1A,E) and p21 (Figure 1A,F) were increased in N27 cells 24 h following MPP^+^ exposure. Intriguingly, the levels of SATB1 were not changed by MPP^+^ treatment (Figure 1A,D). Next, we assessed the effects of α-syn PFF on these same cellular senescence markers using an optimized dosage of α-syn PFF exposure (1 µg/mL) in a time-dependent manner [14,25]. We found that the levels of Lamin B1 (Figure 2A,B), HMGB1 (Figure 2A,C) and SATB1 (Figure 2A,D) were significantly reduced, whereas the level of p21 was increased in N27 cells at 12 h, but not at 48 h following α-syn PFF treatment (Figure 2F). However, p16 was unexpectedly decreased with α-syn PFF (Figure 2A,E). We next quantified the levels of the senescence markers in nuclear and cytosolic fractions of N27 cells (Figure 3A) to verify the expected cellular location of their expressions and specificities. We found that the levels of Lamin B1 (Figure 3A,B), HMGB1 (Figure 3A,C) and SATB1 (Figure 3A,D) were significantly reduced in the nuclear fraction, whereas the level of p21 was increased in the cytosolic fraction by α-syn PFF (Figure 3A,F). Notably, the level of p16 was increased in the nuclear fraction prepared from N27 cells following α-syn PFF treatment (Figure 3A,E).

### 3.2. α-Syn PFF Exposure for the First 48 h Enhances the Expression of p21 in Primary Cortical Neurons

It is still debatable whether external stress induces cellular senescence in neurons or not [5]. Here, we sought to examine the levels of cellular senescence markers in primary cortical neurons following α-syn PFF exposure (5 µg/mL) for cell-type-specific analyses. We found that the levels of Lamin B1 (Figure 4A,B), HMGB1 (Figure 4A,C) and SATB1 (Figure 4A,D) were increased at various time points in primary cortical neurons treated with α-syn PFF. Meanwhile, the level of p21 was increased at 12, 24 and 48 h after α-syn PFF treatment and the level decreased by 72 h (Figure 4A,F). The level of p16 was not significantly changed by α-syn PFF (Figure 4A,E) and the level of β-III-tubulin was decreased with α-syn PFF exposure over time (Figure 4A,G), suggesting the loss of cortical neurons due to α-syn PFF treatment [21].

### 3.3. α-Syn PFF Triggers Cellular Senescence in Astrocytes and Microglia, While It Affects Cortical Neurons Less Drastically

Next, we sought to investigate changes in cellular senescence markers in response to α-syn PFF treatment in primary astrocytes and microglia [21,28]. We found that the levels of Lamin B1 (Figure 5A,B), HMGB1 (Figure 5A,C), SATB1 (Figure 5A,D) and p16 (Figure 5A,E) were significantly reduced, whereas the level of p21 was significantly increased in a time-dependent manner (Figure 5A,F) in primary astrocytes treated with α-syn PFF as assessed by WB analysis. Notably, the level of GFAP was substantially increased, supporting the activation of astrocytes as a consequence of α-syn PFF (Figure 5A,G). Additionally, we quantified the levels of cellular senescence markers in primary microglia with or without α-syn PFF treatment, where we observed mostly similar results to those seen in the treated astrocytes. We found that the levels of Lamin B1 (Figure 6A,B) and p16 (Figure 6A,E) were reduced, whereas the level of p21 was increased (Figure 6A,F) in a time-dependent manner as assessed by WB analysis. The level of HMGB1 was unexpectedly increased in microglia with α-syn PFF (Figure 6A,C). As expected, the level of Iba-1 was significantly increased in a time-dependent manner, supporting the activation of microglia due to α-syn PFF (Figure 6A,G).

### 3.4. The Levels of Cellular Senescence Markers Are Changed in the SNpc of Human PD Patient Postmortem Brains

In the following experiment, we further determined the levels of cellular senescence markers using human PD postmortem midbrain tissues via Western blot analysis (Figure 7A). Our WB analysis revealed that the levels of Lamin B1 (Figure 7A,B), HMGB1 (Figure 7A,C) and SATB1 (Figure 7A,D) were substantially reduced, whereas the p21 level was significantly increased (Figure 7A,F) in the midbrain of human PD postmortem tissues compared to age- and gender-matched controls. This result is similar to our in vitro and ex vivo results, especially with the patterns from primary astrocytes. Notably, p16 levels were not significantly changed in the midbrain of human PD postmortems (Figure 7A,E).

### 3.5. α-Syn PFF Injection Reduces the Levels of Lamin B1 and HMGB1 and Enhances the Levels of p21 and GFAP in the Adult Midbrain and Striatum

We also determined the levels of cellular senescence markers using the ventral midbrain (vMB) and striatum (STR) tissues isolated from the α-syn PFF-injected mouse model of PD via Western blot analysis. We found that the levels of Lamin B1 in the vMB (Figure 8A,B) and the striatum (Figure 8J,K), HMGB1 in the vMB (Figure 8A,C) and the STR (Figure 8J,L) and p16 in the vMB (Figure 8A,E) and the STR (Figure 8J,N) were reduced, whereas the levels of p21in the vMB (Figure 8A,F) and the STR (Figure 8J,O) were substantially increased in the α-syn PFF-injected mice. However, we did not find consistent changes in SATB1 in the vMB (Figure 8A,D) and the STR (Figure 8J,M). We also confirmed that the levels of GFAP were increased in the vMB (Figure 8A,G) and the STR (Figure 8J,P) as well as the levels of Iba-1 in the vMB (Figure 8A,H) and the STR (Figure 8J,Q), which suggest that both astrocytes and microglia are activated by α-syn PFF injection. However, the levels of β-III-tubulin were reduced in the vMB (Figure 8A,I) and the STR (Figure 8J,R) due to α-syn PFF injection. This result is consistent with our previous findings [21].

### 3.6. In Vivo α-Syn PFF Inoculation Leads to Microglia Activation and Reactive Astrocytes in the STR, SNc and CTX

Next, we employed immunohistochemistry (IHC)-based analysis to determine the changes on cellular senescence markers in specific cell-types in α-syn PFF-injected mice. To this end, the STR, SNc and CTX tissues were used to capture IHC images using confocal microscopy and EVOS (Thermo) imaging system. Since the levels of Lamin B1, HMGB1 and p21 were consistent in the results above, we focused on assessing those markers in astrocytes and microglia, without including the analysis of cortical neurons. In Figure 9, Figure 10 and Figure 11, we demonstrate the immunoreactivity of Lamin B1, HMGB1 and p21 in astrocytes and microglia with cell type-specific markers: GFAP for astrocytes and Iba-1 for microglia in the STR (Figure 9), SNc (Figure 10) and CTX (Figure 11). The analyses revealed that the levels of Lamin B1 and HMGB1 were significantly decreased, whereas the level of p21 was increased in the STR (Figure 9), SNc (Figure 10) and CTX (Figure 11). The IHC analysis for assessing co-localization with cell-type markers revealed that the immunoreactivity of β-III tubulin-positive neurons in the SNc was also reduced by α-syn PFF injection (Appendix A), whereas the label intensities of GFAP-positive astrocytes and Iba-1-postivie microglia were increased by α-syn PFF (Figure 9, Figure 10 and Figure 11). In addition, we found that the numbers of GFAP-positive astrocytes in the STR (Figure 12A,B), SNc (Figure 12A,D) and CTX (Figure 12A,F) were increased by α-syn PFF inoculation. Furthermore, the soma volumes of astrocytes in the STR (Figure 12A,C), SNc (Figure 12A,E) and CTX (Figure 12A,G) were increased as assessed by the outlined GFAP-positive cell body as previously described [21], suggesting that both astrocytes and microglia are activated by α-syn PFF. Meanwhile, β-III-tubulin-positive neurons were reduced by α-syn PFF inoculation in the STR, SNc (Appendix A) and CTX, supporting neuronal loss.

## 4. Discussion

The objective of this study was to determine whether PD pathology mimicking α-syn PFF results in the induction of cellular senescence in various CNS cell-types including neurons, microglia and astrocytes as a mechanism of PD pathology. We demonstrate in our studies that both MPP^+^ and α-syn PFF-induced stress resulted in changes in cellular senescence markers including decreases in Lamin B1 and HMGB1 and increases in p21 in dopaminergic N27 cells, forebrain-derived primary astrocytes and microglia, and in mouse STR, SNc and cortex, which are consistent with the patterns observed in the SNpc of human PD patients. The only exception was that HMGB1 was increased with α-syn PFF in primary cortical microglia. Our human PD results are supportive of a recent report, demonstrating that the levels of p21 are higher in human PD SNpc than age-matched controls [20]. Interestingly, our results also show that neurons, at least cortically derived, do not demonstrate a reduction in Lamin B1 and HMGB1; however, the up-regulation of p21 was observed for the first 48 h following α-syn PFF treatment, even though the levels of β-III-tubulin declined due to α-syn PFF toxicity [16,20,33]. Cortical neurons lacking SATB1 do not show a typical senescence phenotype, which is somewhat aligned with our results [20]. Therefore, our results suggest that α-syn PFF-induced cellular senescence is more likely to be mediated by astrocytes and microglia rather than neurons [15,21].

We also found that some senescence markers may have a critical expression time-window, especially in neurons. For example, p21 expression was elevated for the first 48 h following α-syn PFF treatment but declined after 48 h of α-syn PFF exposure in both dopaminergic N27 cells and primary cortical neurons, which was not the case in astrocytes and microglia (Figure 2 and Figure 4, Figure 5 and Figure 6). However, the expression of typical cellular senescence markers (Lamin B1 and HMGB1) gradually declined over time in primary astrocytes and microglia isolated from the forebrain, which was validated via immunohistochemistry in the striatum, SNc and cortex of α-syn PFF injected mice (Figure 9, Figure 10 and Figure 11). When mouse brains were analyzed 5–6 months after PFF injection, both Lamin B1 [20,22] and HMGB1 [16] were consistently reduced in the striatum and midbrain, and p21 [20] was substantially enhanced in both regions in Western blot analyses (Figure 8). These results are well aligned with the patterns we observed in post-mortem human PD SNpc tissues. We verified these findings via IHC, demonstrating that α-syn PFF injection was found to coincide with a significant reduction in the levels of Lamin B1 and HMGB1 within astrocytes and microglia located in the striatum, SNc and cortex, although the expression levels of GFAP and Iba-1 as well as the numbers of GFAP- or Iba-1-positive cells were substantially increased in the α-syn PFF-injected mouse brains [8,21]. This result was found to coincide with increased p21 levels in those same cells in brains [8,16,19]. In contrast, our IHC analysis showed the reduced levels of Lamin B1 and HMGB1 in neurons, which may be undermined due to the neuronal loss derived from α-syn PFF. Importantly, the soma volume of astrocytes was also substantially enhanced, supporting that α-syn PFF toxicity may activate reactive astrocytes (and perhaps damaged microglia) as a part of its toxic mechanism [8,21]. In addition, our results strongly suggest that the overall decreases in Lamin B1 and HMGB1 and an increase in p21 may be derived from astrocytic and/or microglial senescence, which were induced by α-syn PFF treatment. Our combined results suggest that both senescent and reactive astrocytes may play a critical role in mediating protein-aggregation based neuronal loss in PD pathology [21,22,34]. As a follow-up, it would be interesting to distinguish non-proliferative senescent astrocytes/microglia from proliferative reactive astrocytes or microglia [5]; further, it can be an intriguing but promising approach to apply senolytics to the α-syn PFF injected mouse model to see whether the pharmacological intervention would be sufficient to halt or reverse Lewy-body like protein aggregation-induced PD pathology, since its possibility was suggested in tau-dependent pathology [22,23,24] as well as renal regeneration [35].

Since our focus was not to assess the effects on SASP by α-syn PFF, we did not assess all the SASP-related cytokines, but we found in qRT-PCR that IL-8 was up-regulated by α-syn PFF exposure in both cortex-derived astrocytes and microglia culture (data not shown), as reported in human PD post-mortem SNpc [7]. We also recognize some discrepancies between our results and a few previous reports that other senescence markers, such as SATB1 and LAMP1 were down-regulated and p16 was up-regulated by the external stress [7,20]. We speculate that some discrepancies can be explained by the time point of measuring those factors, as well as the types of external stressor and responding cells [22]. Although both p21 and p16 are cyclin-dependent kinase inhibitors to arrest cell cycle, p21 tends to be up-regulated in early stage of senescence, which was probably detected in our in vitro and ex vivo results, but p16 levels can be consistently high even in late stage of senescence [36]. It will be an intriguing study to distinguish p21- vs. p16-dependent senescence pathway as a follow-up study. Taken together, our results support the potential significance of targeting “astrosenescence” as a novel approach to prevent aging-related neurodegenerative diseases, including PD [15].

## 5. Conclusions

Our results strongly suggest that α-syn PFF-induced toxicity reduces the levels of cellular senescence markers, such as Lamin B1 and HMGB1 in the midbrain and striatum, whereas it increases the level of p21 through senescent astrocytes and microglia.

## Figures and Tables

**Figure 1 cells-10-01694-f001:**
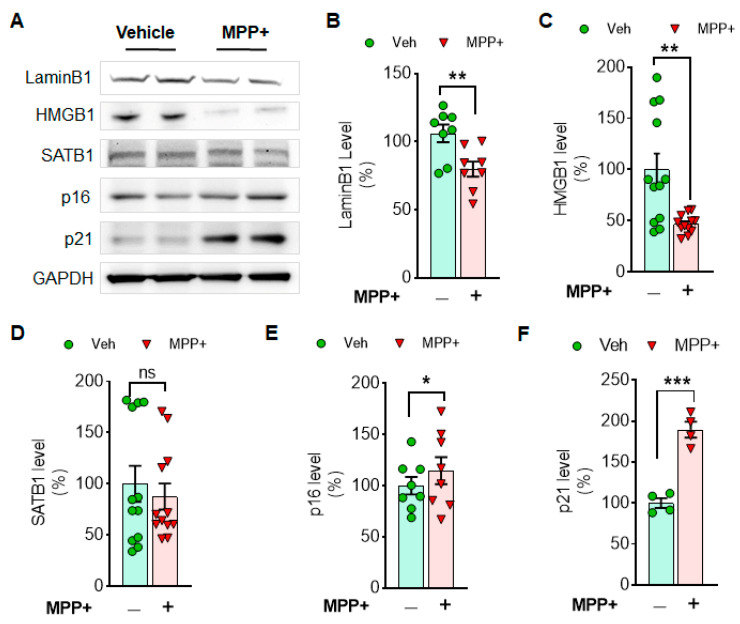
MPP^+^ toxicity reduces the levels of Lamin B1 and HMGB1, while it increases a cell arrester, p21 in N27 cells. The examples of cellular senescence markers are displayed in Western blots (**A**). The levels of LaminB1 (**B**), HMGB1 (**C**), SATB1 (**D**), p16 (**E**) and p21 (**F**) are displayed in comparison between MPP^+^ and vehicle treatment. Cellular senescence markers such as LaminB1 and HMGB1 were reduced, but p16 and p21 were enhanced by MPP^+^ treatment. GAPDH was adopted as a loading control. ImageJ was used for the analyses of band intensities, and relative levels (100% for vehicle) are displayed in mean ± SEM and applied to unpaired Student’s *t*-test for statistical significance. *: *p* < 0.05, **: *p* < 0.01 and ***: *p* < 0.001. ns: not significant.

**Figure 2 cells-10-01694-f002:**
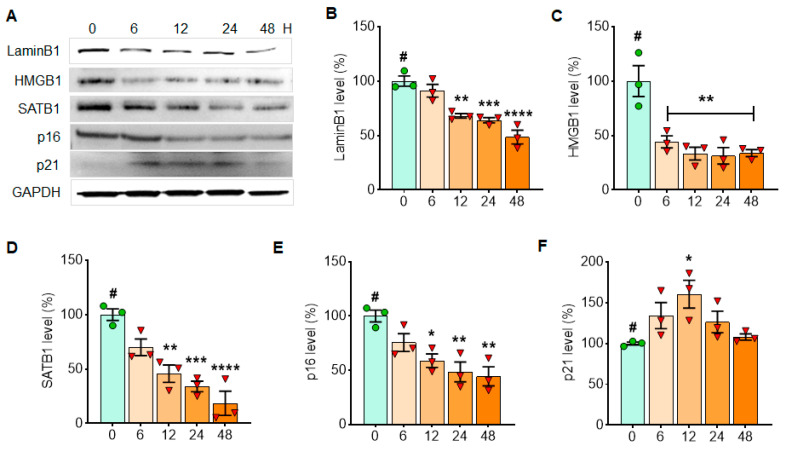
α-Syn PFF treatment reduces the expressions of Lamin B1, HMGB1 and SATB1 in N27 cells, whereas it increases p21 expression in a short period of time. The examples of cellular senescence markers over 48 h are displayed in Western blots (**A**). The levels of LaminB1 (**B**), HMGB1 (**C**), SATB1 (**D**), p16 (**E**) and p21 (**F**) with α-syn PFF treatment are displayed in comparison with initial time point (T = 0). GAPDH was adopted as a loading control, and ImageJ was used for the analyses of band intensities. The relative levels (100% for no PFF treated vehicle) are displayed in mean ± SEM and applied to one-way ANOVA, Dunnett’s post-hoc test (#: compared with T = 0 point) for statistical significance. **: p* < 0.05, ***: p* < 0.01, ****: p* < 0.001 and *****: p* < 0.0001.

**Figure 3 cells-10-01694-f003:**
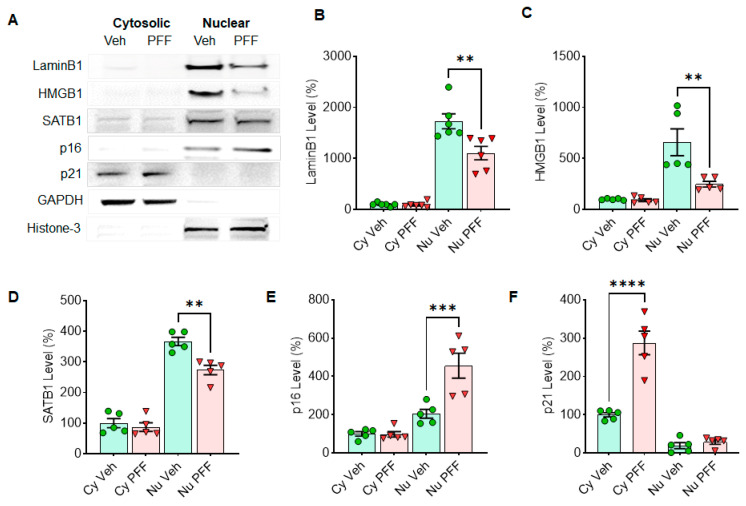
Senescence markers in N27 cells are regulated in nuclear (Nu) or cytosolic (Cy) fraction by α-syn PFF treatment. The α-syn PFF treatment reduces the levels of Lamin B1, HMGB1 and SATB1 in nuclear fractions, whereas it increases the levels of p16 in nuclear and p21 in cytosolic fractions after 48 h of exposure. The examples of cellular senescence markers are displayed in Western blots (**A**). The band intensities of Lamin B1 (**B**), HMGB1 (**C**), SATB1 (**D**), p16 (**E**) and p21 (**F**) were quantified using ImageJ. GAPDH and histone-3 were used as a cytosolic- and a nuclear-loading control, respectively. The relative levels (100% for vehicle treatment) are displayed in mean ± SEM and applied to unpaired Student’s *t*-test for statistical significance. ***: p* < 0.01, ****: p* < 0.001 and *****: p* < 0.0001.

**Figure 4 cells-10-01694-f004:**
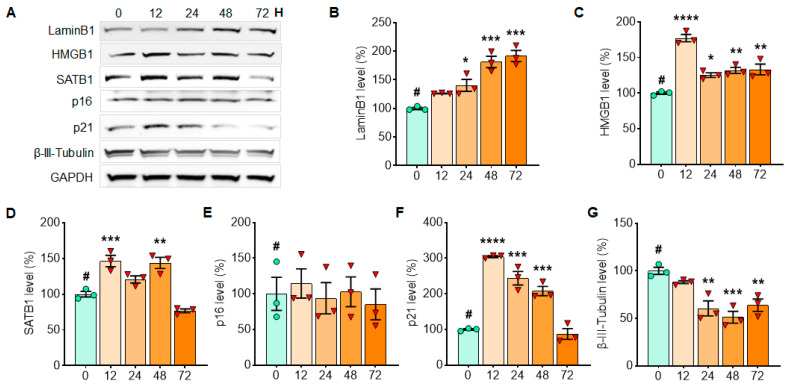
The level of p21 was enhanced by α-syn PFF exposure in primary cortical neuronal culture for the first 48 h and then it declined gradually. In Western blots, the levels of senescence markers were quantified in the primary cortical neuron at 12, 24, 48 and 72 h after α-syn PFF exposure to detect the expression patterns over time (**A**). In the analyses of the expression patterns over 3 days of α-syn PFF exposure, the levels of Lamin B1 (**B**), HMGB1 (**C**), SATB1 (**D**) and p21 (**F**) increased but β-III-tubulin (**G**) declined over time; however, p16 (**E**) was not changed significantly. GAPDH was adopted as a loading control, and ImageJ was used for the analyses of band intensities. The relative levels (100% for no PFF treatment) are displayed in mean ± SEM and applied to one-way ANOVA, Dunnett’s post-hoc test (#: compared with T = 0 point) for statistical significance. **: p* < 0.05, ***: p* < 0.01, ****: p* < 0.001 and *****: p* < 0.0001.

**Figure 5 cells-10-01694-f005:**
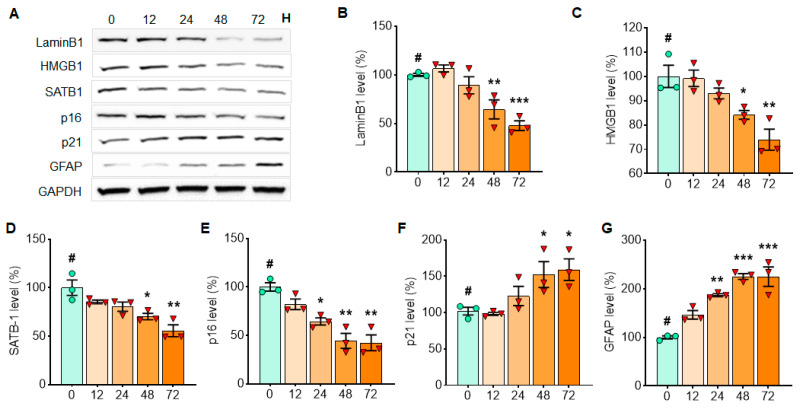
The levels of cellular senescence markers, such as LaminB1, HMGB1, SATB1 and p16, gradually decrease with α-syn PFF exposure, while the levels of p21 and GFAP increase in the isolated astrocyte culture over 3 days with α-syn PFF. In Western blots, cellular senescence markers were quantified in primary astrocytes culture at 12, 24, 48 and 72 h after α-syn PFF treatment to detect expression patterns over time (**A**). For assessing the expression patterns over 3 days of α-syn PFF exposure, we quantified the levels of Lamin B1 (**B**), HMGB1 (**C**), SATB1 (**D**), p16 (**E**), p21 (**F**) and GFAP (**G**). GAPDH was used as a loading control. The relative band intensities (100% for no PFF treatment) are displayed in mean ± SEM and applied to one-way ANOVA, Dunnett’s post-hoc test (#: compared with T = 0 point) for statistical significance. **: p* <0.05, ***: p* < 0.01 and ****: p* < 0.001.

**Figure 6 cells-10-01694-f006:**
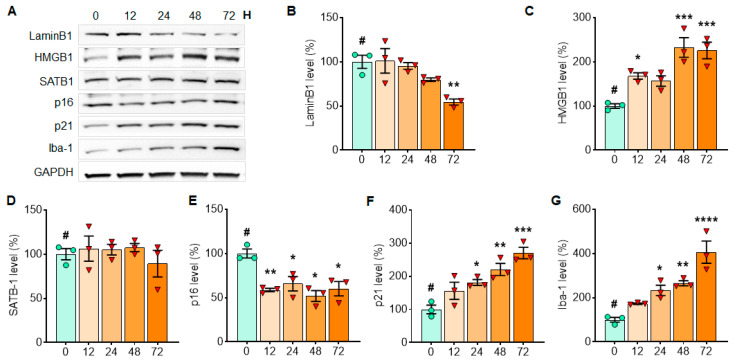
The levels of LaminB1 and p16 gradually decrease with α-syn PFF treatment, while the levels of HMGB1, p21 and Iba-1 increase in the isolated microglial culture over 3 days of α-syn PFF exposure. In Western blots, senescence markers were quantified in primary microglia culture at 12, 24, 48 and 72 h after α-syn PFF exposure to detect expression patterns over time (**A**). For assessing the expression levels over 3 days of α-syn PFF exposure, we quantified the levels of Lamin B1 (**B**), HMGB1 (**C**), SATB1 (**D**), p16 (**E**), p21 (**F**) and Iba-1 (**G**). GAPDH was also used as a loading control. The relative band intensities (100% for no PFF treatment) are displayed in mean ± SEM and applied to one-way ANOVA, Dunnett’s post-hoc test (#: compared with T = 0 point) for statistical significance. **: p* < 0.05, ***: p* < 0.01, ****: p* < 0.001 and *****: p* < 0.0001.

**Figure 7 cells-10-01694-f007:**
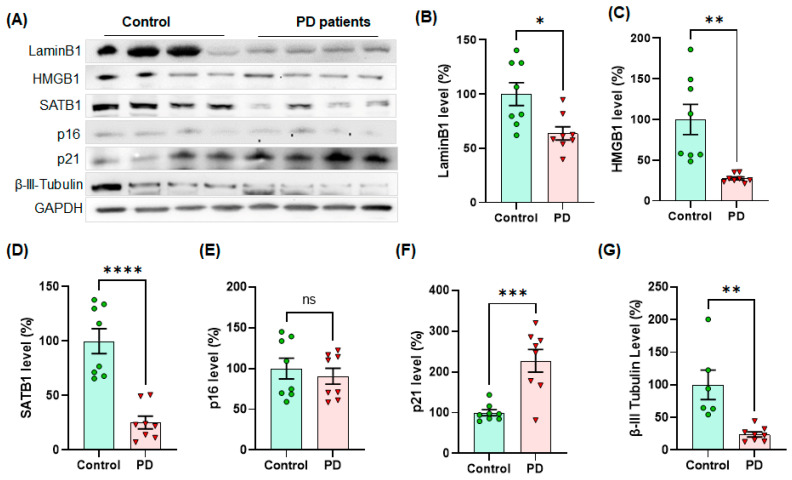
The levels of cellular senescence markers, such as Lamin B1, HMGB1 and SATB1, were significantly lower in human PD SNpc than age- and gender-matched control tissues, whereas the level of p21 was higher in human PD post-mortem SNpc (n = 8/group). In Western blots, the examples of senescence markers are displayed in quadruplets per group (**A**). The quantified levels of Lamin B1 (**B**), HMGB1 (**C**), SATB1 (**D**), p16 (**E**), p21 (**F**) and β-III-tubulin (**G**) were statistically analyzed in unpaired Student’s *t*-test for significance. The band intensity of GAPDH was normalized and displayed as relative band intensities (100% for age-matched controls, n = 8/group) in mean ± SEM. **: p* < 0.05, ***: p* < 0.01, ****: p* < 0.001 and *****: p* < 0.0001. ns: not significant.

**Figure 8 cells-10-01694-f008:**
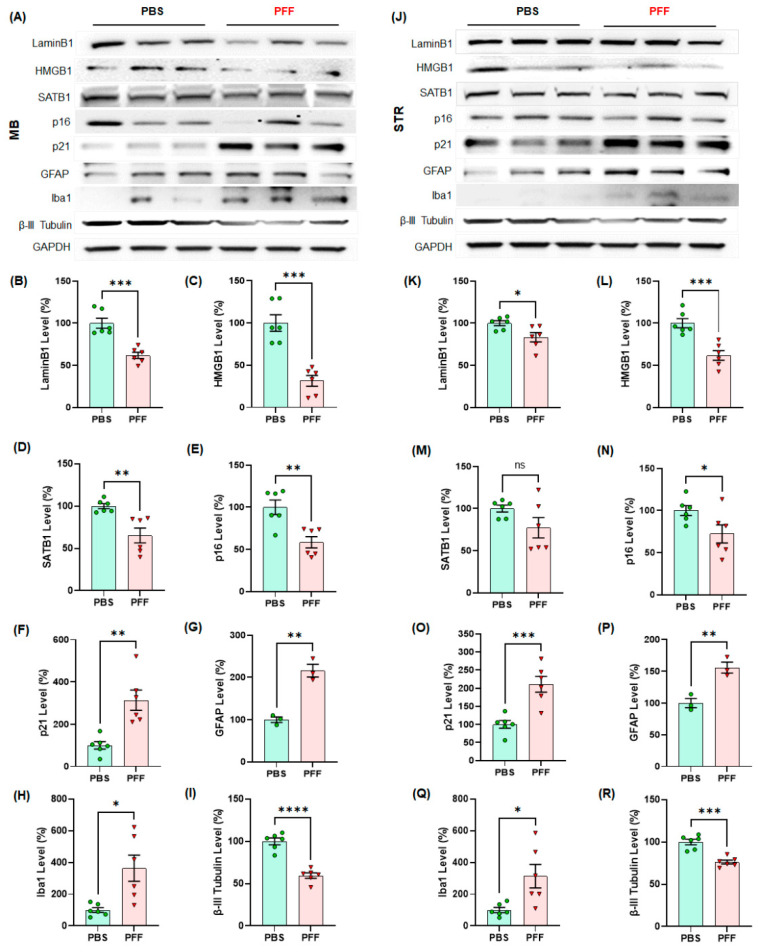
The levels of cellular senescence markers such as LaminB1 and HMGB1 were significantly lower in α-syn PFF-injected mouse midbrain (MB) and striatum (STR) than PBS-treated brains, whereas the levels of p21, GFAP and Iba-1 were enhanced by 5–6 months after PFF treatment (n = 6/group). In Western blots, the examples of cellular senescence markers are displayed in triplicates per group ((**A**) MB, (**J**) STR). The quantified levels of Lamin B1 (**B**,**K**), HMGB1 (**C**,**L**), SATB1 (**D**,**M**), p16 (**E**,**N**), p21 (**F**,**O**), GFAP (**G**,**P**), Iba-1 (**H**,**Q**) and β-III-tubulin (**I**,**R**) are demonstrated. The level of β-III-tubulin decreased with α-syn PFF injection due to the neuronal loss. In quantification, the band intensity was normalized by a loading control, GAPDH and displayed in mean ± SEM in relativity (100% for age-matched controls, n = 6/group). The data analysis was applied to unpaired Student’s *t*-test for statistical significance. **: p* < 0.05, ***: p* < 0.01, ****: p* < 0.001 and *****: p* < 0.0001. ns: not significant.

**Figure 9 cells-10-01694-f009:**
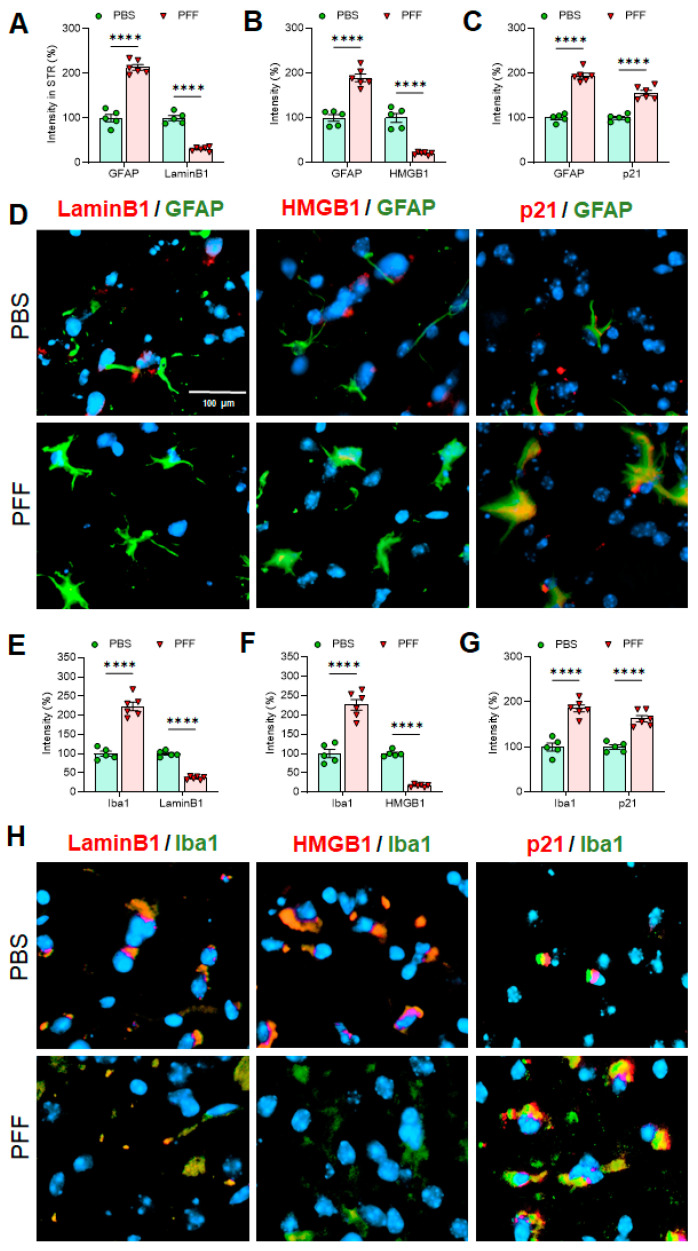
In cell-type-specific analyses in the STR, the levels of Lamin B1 and HMGB1 are significantly lower in both astrocytes and microglia in α-syn PFF-injected mouse brains than PBS-treated ones, whereas the levels of p21 in both cell-types are increased with α-syn PFF. In the STR, both GFAP (**A**–**C**) and Iba-1 (**E**–**G**) were increased in PFF-injected STR (n = 6/group). The intensities of Lamin B1 and HMGB1 in both astrocytes (**A**,**B**,**D**) and microglia **(E**,**F**,**H**) were reduced, while the levels of p21 were significantly higher with α-syn PFF (**C**,**D**,**G**,**H**). The label intensities of Lamin B1, HMGB1 and p21 were quantified cell-type specifically (GFAP or Iba-1), based on double-labels in a blinded manner. DAPI stain (blue) was used to indicate the location of nucleus. In quantification, PBS injected STR region was used as a relative label intensity (100%, n = 5–6/group) in mean ± SEM and applied to unpaired Student’s *t*-test for statistical significance. *****: p* < 0.0001. Size bar: 100 µm.

**Figure 10 cells-10-01694-f010:**
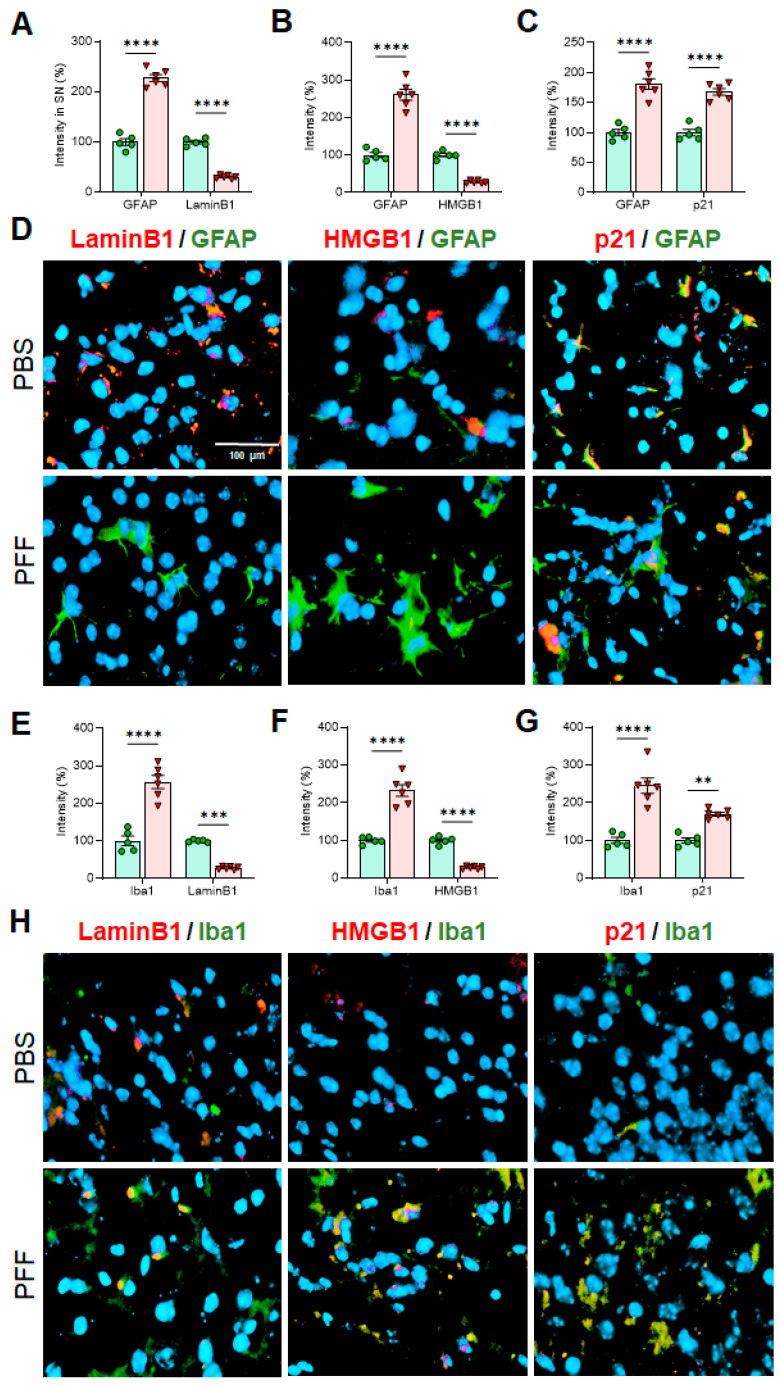
In cell-type-specific analyses in the SNc, the levels of Lamin B1 and HMGB1 are significantly decreased in both astrocytes and microglia in α-syn PFF-injected mouse brains than PBS-treated ones, whereas the levels of p21 in both cell types are increased with α-syn PFF. In the SNc, both GFAP (**A**–**C**) and Iba-1 (**E**–**G**) label intensities were increased in PFF-injected SNc (n = 6/group) than PBS control. The intensities of Lamin B1 and HMGB1 in both astrocytes (**A**,**B**,**D**) and microglia (**E**,**F**,**H**) were reduced; however, the levels of p21 were significantly increased in both cell-types with α-syn PFF (**C**,**D**,**G**,**H**). DAPI stain (blue) was used to indicate the location of nucleus. The label intensities of Lamin B1, HMGB1 and p21 were cell-type specifically quantified, based on double-labels in a blinded manner. In quantification, PBS injected SNc region was used as a relative label intensity (100%, n = 5/group) in mean ± SEM and applied to unpaired Student’s *t*-test for statistical significance. ***: p* < 0.01, ****: p* < 0.001 and *****: p* < 0.0001. Size bar: 100 µm.

**Figure 11 cells-10-01694-f011:**
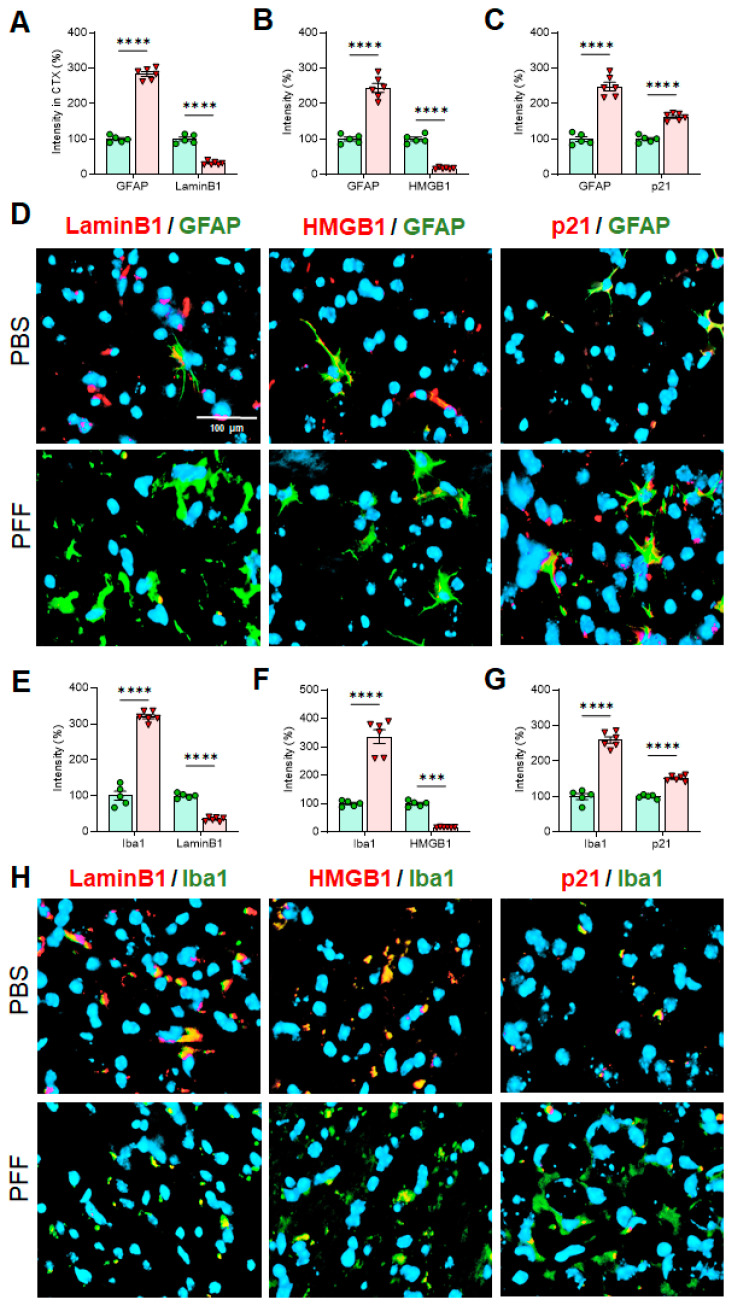
In cell-type-specific analyses in the CTX, the levels of Lamin B1 and HMGB1 are significantly lower in both astrocytes and microglia in α-syn PFF-injected brains than PBS-treated ones, whereas the levels of p21 in both cell types are increased with α-syn PFF. In the CTX, both GFAP (**A**–**C**) and Iba-1 (**E**–**G**) were increased in PFF-injected CTX (n = 6/group). The intensities of Lamin B1 and HMGB1 in both astrocytes (**A**,**B**,**D**) and microglia (**E**,**F**,**H**) were reduced; however, the levels of p21 were enhanced in both cell types with α-syn PFF (**C**,**D**,**G**,**H**). DAPI stain (blue) was also used to indicate the location of nucleus. The label intensities of Lamin B1, HMGB1 and p21 were quantified cell-type specifically, based on double-labels in a blinded manner. In quantification, PBS injected CTX region was used as a relative label intensity (100%, n = 5/group) in mean ± SEM and applied to unpaired Student’s *t*-test for statistical significance. ****: p* < 0.001 and *****: p* < 0.0001. Size bar: 100 µm.

**Figure 12 cells-10-01694-f012:**
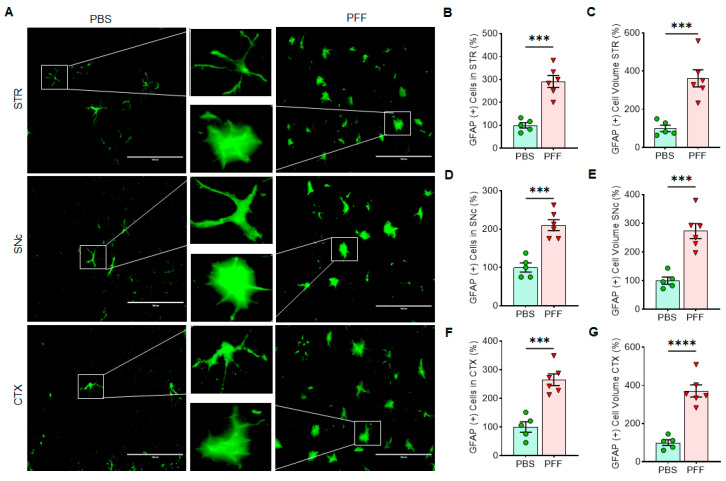
The number of GFAP-positive astrocytes as well as the cell volume of astrocytes were significantly increased with α-syn PFF treatment in the STR, SNc and CTX. (**A**) The examples of enlarged astrocytes by α-syn PFF are compared with PBS treated controls and displayed in the STR, SNc and CTX. The astrocyte counting (**B**,**D**,**F**) and cell volume measurement (**C**,**E**,**G**) were analyzed in a blinded manner. In quantification, α-syn PFF-injected brains (n = 6/group) were compared with PBS-injected brains (n = 5/group) in the STR (**B**,**C**), SNc (**D**,**E**) and CTX (**F**,**G**), displayed as a relative number or volume (100% in PBS) in mean ± SEM and applied to unpaired Student’s *t*-test for statistical significance. ***: *p* < 0.001 and ****: *p* < 0.0001. Size bar: 100 µm.

## Data Availability

All data generated or analyzed for the study are included in this published article (and its Appendix A).

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
