# Peer review of "Alpha-Synuclein Preformed Fibrils Induce Cellular Senescence in Parkinson’s Disease Models"

_cells, 2021, doi:10.3390/cells10071694_

Round 1
Reviewer 1 Report
The paper by Kumar Verma et al., describes the effects on cellular senescence induced by treatment with alpha-synuclein preformed fibrils in N27 cells, primary cortical cell cultures, mice and human. Although the paper shows some interesting results, the conclusions drawled by authors are not well supported by the results. Besides, the paper has several important issues that should be addressed before it will be suitable for publication.
- Authors should better explain why they have chosen LaminB1, HMGB1, p16 and p21 as senescence markers, supporting this selection by some references. Which is the function of these proteins? Why do they change with senescence?
- They also should include a more complete discussion about the relationship of aging and Parkinson´s disease (e.g. PMID: 314287316; PMID: 25261764; PMID: 25261764).
- In Fig.1 the bands for SATB1 are very weak in comparison with the rest of the figures. Authors should improve this picture.
- In the figure legends authors remark only some of the results showed in the figures (for example, in figure 1 they explain results for LaminB1, HMGB1 and p21 but nothing about p16; in figure 4 they only explain the results regarding P21, etc). I suggest being more consistent with the way that they presents the results and figures.
- Title of 3.2 heading (α-Syn PFF limits the changes on cellular senescence makers in primary cortical neurons) is confusing.
- Authors should also explain why they have used animals of 2, 8 and 12 month-old in the study.
- Abbreviations should be defined the first time they appear and then be consistent throughout the text (e.g. cortex, CTX).
- The manuscript has some grammatical and spelling mistakes (for example, line 225 “makers” instead of “markers; line 149, or line 360)
- Line 219 in figure legend 3 “whereas it increases the p21...” should be changed by “whereas it increases the p21 and p16 levels in cytosolic fraction after 48 h of exposure”.
- I miss an explanation about why authors measure the levels of senescence markers on the nuclear and cytosolic fraction only in N27 cells and which is the meaning of the different results obtained in this experiment.
- In line 249 authors state that “we observe similar results to those seen in the treated astrocytes”. However, results regarding HMGB1 and SATB1 are not similar.
- In line 277 authors state “This result is similar to our in vitro and ex vivo results”. If the ex vivo experiment are those using primary cell cultures, results with human samples are not similar .
- I have understood that the in vivo experiment consist in injecting alpha-synuclein fibrils in the striatum, wait for 5-6 months and then study the SN, striatum and cortex. If this is correct it should be better explain in material and methods section.
- 2.8 and 2.9 heading are quite similar and confusing. Did authors insert a cannula unilaterally or bilaterally? Are these animals used for different techniques? Why in 2.8 they use 2 month-old animals and in 2.9 8-12 month-old animals?
- Figure 8. Why authors have measured GFAP in striatum and midbrain but they have not measure microglia in these structures using Iba-1?
- Authors should show data for beta-tubulin-positive neurons (line 316).
- Immunofluorescence figures are very small and do not allow to see the quality of the staining.
- It is not clear if the results shown in figure A are expressed in number/amount of astrocytes/microglia that colocalize with LaminB1, HMGB1 or p21. The method to perform this quantification should be described. Why astrocytes are counted by two different methods (GFAP level (%) and GFAP+cells (%))? Did authors actually count the number of astrocytes/mm2 or it is a measure of the signal intensity of this protein?
- Lines 361-364, authors state that “We demonstrate in our studies that both MPP+ and α-syn PFF-induced stress resulted in changes in cellular senescence markers including decreases in Lamin B1 and HMGB1 and increases in p21 in dopaminergic N27 cells, forebrain-derived primary astrocytes and microglia, and in mouse STR, SNc and cortex, which are consistent with the patterns observed in the SNpc of human PD patients”. However, this is not correct in some of the experiments performed in this study. In some cases senescence markers are increased while in other cases they decrease.
- Line 368, are authors describing their results or citing previous results? This is not clear. Same comment for line 378.
- I think is better not to include references to the figures in the Discussion Section.
- In this paper authors has deep into the changes of astrocytes in this experimental model. Why not do it also with the microglia? There are easy techniques to study the different phenotypes of microglial cells.
- At the end of the discussion authors state that they have measured the levels of some SASP such as IL-8, but they do not show these results. Why? The paper would strengthen including IL-8 measurement along with other SASP such as IL-6.
- In line 386 authors state that “Our results strongly suggest that reactive astrocytes play a critical role in mediating protein-aggregation based neuronal loss in PD pathology”. This is also the conclusion of the paper. Why astrocytes and not microglial cells, that are also affected by alpha-syn treatment in the same way that astrocytes?
Author Response
Thank you for your comments on this manuscript, please see the attachment for the author’s reply

Reviewer 2 Report
The manuscript is well writed and with well english. Authors try to demonstrate the involvement of cellular senescence in Parkinson’s disease.
They found that different senescence makers within reactive astrocytes are rerelated to a enlarged cell bodies and GFAP-positive cell and Iba1-activated microglia in α-syn PFF injected mouse brains. They indicate that pathology could lead to astrocytes and/or microglia senescence in PD brain and perhaps may contribute to neuropathology.
Although their results show interesting findings in the field, following issues need to be addressed.
Major revision
1.- They do not used RT-PCR that probably will help them to obtain better results.
2.- Astrocytes are so important cells and microglia too, but dendrites and synapses degenerate and neuron releases vimentin in an attempt to re-grow the dentrite tree branches and synapses. Authors need to do experiments to detect vimentin increase or decrease in their model. Also authors need to discuss about that. If neurons liberate growth factors to protect senescence of astrocytes and/or microglia need to be discuss. Perhaps a mixer culture of astrocytes and neurons, or microglia and neurons or all together will be interested.
3.- Why have measured these proteins and not others? You measured GFAP but what happen with neurons, can you measure perhaps map-2 or other neuron marker?
4.- Authors need to explain better the differences of p16 and p21. What are they expecting? What they obtain? Why they think other authors published another results.
5.- There are not differences in some western-blots and sometimes graphs are not in concordance with the photograph presented.
6- Authors have not discussed enough about the state of the art and the different conclusions by other authors. Have other authors obtained similar results? and have other authors differents results and why?
7.- Can authors add bibliography? they are not enough.
Minor revision
- There are mistakes in the paper
- Spaces bad included, etc.
Author Response
Thank you for your comments on this manuscript, please see the attachment for the author’s reply.

Reviewer 3 Report
The authors gauge the extent of senescence in several Parkinson’s disease-related models, and human brain tissue, using a number of senescence-linked markers, as well as cell markers. As the authors point out, this is an important area of study given removal of fibrils still leaves cells unhappy, suggesting extra therapy beyond proteinopathy-focussed treatments might be needed.
Study had good breadth - examining the research question across in vitro, in situ and in vivo setups. It also looked at multiple species. A number of relevant markers were used and a number of cell types were examined. The paper was also well written and the immunofluorescence images were of a high quality.
It is the opinion of this reviewer that the work would benefit from addressing the following points:
- Was a thioflavin T assay done to confirm the fibrillar nature of the PFFs?
- Classic full examination of senescence would include assessment of SA-b-gal activity and demonstrating a senescence-associated secretory profile (SASP). Although the Discussion mentioned that showing a SASP wasn’t a part of the research question, demonstration of SASP is usually thought of as core to defining cells as senescent. A separate paper is referenced to show one cytokine is elevated, but it would be good to examine some of the core SASP cytokines (eg IL-6) and look at SA-b-gal activity on samples in which this is easy to do (eg N27 cells, primary astrocytes and microglia).
- A number of markers (eg lamin B1, HMGB1, SATB1 and surprisingly p16) were reduced after alpha-synuclein PFF treatment of N27 cells - were the cells alive at the end of the treatment (ie could this decrease have been due to cell death)?
- Figure 3 - add a definition of ‘Cy’ and ‘Nu’ to the figure legend
Author Response

(The authors gave the same response as above.)

Round 2
Reviewer 1 Report
Authors have answer all my questions and amended all the recommendations I have made in my reviewer report.
Reviewer 2 Report
Accept in present form